# A theoretical morphological model for quantitative description of the three-dimensional floral morphology in water lily (*Nymphaea*)

**Shiryu Kirie[1], Hideo Iwasaki[2], Koji Noshita[3,4,5], Hiroyoshi Iwata**[1] *

**1** Department of Agricultural and Environmental Biology, Graduate School of Agricultural and Life Sciences, University of Tokyo, Bunkyo, Tokyo, Japan, **2** metaPhorest, Department of Electrical Engineering and Bioscience, Waseda University, TWIns, Shinjuku, Tokyo, Japan, **3** Department of Biology, Faculty of Science of Science, Kyushu University, Fukuoka, Fukuoka, Japan, **4** Plant Frontier Research Center, Kyushu University, Fukuoka, Fukuoka, Japan, **5** Japan Science and Technology Agency, PRESTO, Kawaguchi, Saitama, Japan

* hiroiwata@g.ecc.u-tokyo.ac.jp

**Data Availability Statement:** The numerical files are available from https://github.com/ShiryuKirie/theoretical-morphological-model-of-water-lily.

## Abstract

Water lilies (N*ymphaea* spp.) have diverse floral morphologies. Water lilies are not only commonly used as ornamental plants, but they are also important for understanding the diversification of basal angiosperms. Although the diversity in floral morphology of water lily provides useful information for evolutionary biology, horticulture, and horticultural science, it is difficult to describe and analyze the three-dimensional morphology of flowers. In this study, we propose a method to describe the floral morphology of water lily using a three-dimensional theoretical morphological model. The theoretical model was constructed based on three components, i.e., (1) the gradual change in size of floral organs, (2) spiral phyllo-taxis, and (3) the interpolation of elevation angles, which were integrated into the model. We generated three-dimensional representation of water lily flowers and visualized theoretical morphospaces by varying each morphological parameter. The theoretical morphospace is a mathematical space of morphological spectrum generated by a theoretical morphological model. These morphospaces seems to display the large part of morphological variations of water lily. We measured morphological parameters of real flowers based on our theoretical model and display the occupation pattern of morphological parameters. We also surveyed the relation between morphological parameters and flower shape descriptions found in a catalog. In some parameters, we found breeders' description can link to our morphological model. In addition, the relationship between the global features of floral morphology and the parameters of the theoretical model was calculated with flower silhouettes simulated with a range of parameter values and the global features of the silhouette. We used two simple indices to assess the global morphological features, which were calculated with the convex hull. The results indicated that our method can effectively provide an objective and quantitative overview of the diversity in the floral morphology of water lily.

**Funding:** This study was partially supported by JST CREST Grant Number JPMJCR16O2. The funder had no role in study design, data collection and analysis, decision to publish, or preparation of the manuscript.

**Competing interests:** The authors have declared that no competing interests exist.

## Introduction

Diversity in flowering plants has been an important subject in plant evolution, also referred to as Darwin's "abominable mystery" [1, 2]. Approximately three million species of angiosperms have been recognized till date [3]. Flowers of angiosperms show diverse morphological variations, which might have been influenced by interactions with other organisms, e.g., interaction with a pollinator, also known as pollination syndrome [4–8]. The diversity in floral morphology of angiosperms has been enhanced not only by natural selection, but also through artificial selection. Plant breeders involved in floriculture have developed numerous cultivars, with novel morphological features and high aesthetic values, for use as ornamental plants. The morphological changes and diversification resulting from floricultural breeding are referred to as domestication syndrome [9]. In some ornamental plants, such as chrysanthemum and morning-glory in Japan, diversity in floral morphology has resulted from historical cultural movements [10]. Thus, variations in floral morphology of ornamental plants may also be important for studying their artistic and cultural contributions.

As described above, floral morphology is a subject of interest in various research fields, including evolution, ecology, genetics, breeding, and art. To facilitate studies on floral morphology, it is desirable to have models and methods for quantitatively describing morphological variations in flowers, because quantitative description of floral morphology yields more information than qualitative description, and may thus allow identification of variations caused by natural and artificial selection.

Quantitative descriptions of floral morphology have not been studied extensively. In particular, three-dimensional floral morphology has not been studied sufficiently because of its complexity resulting from the different types of organs involved, such as petals, calyces, pistils, and stamens. In *Primula sieboldii*, an elliptic Fourier descriptor (EFD) [11] was used to estimate the shape of petals [9, 12, 13]. Yoshioka *et al.* [9] compared the Japanese wild population and traditional cultivars of *P. sieboldii* and evaluated the change and diversification in the petal shape and size through its breeding history in Japan. In *Eustoma grandiflorum*, the relationship between the petal and corolla silhouette shapes was analyzed by combining EFD and principal component analysis (PCA) [14–16]. In this species, developmental changes in the curvature of the petal and the association between petal curvature and corolla silhouette shape were also investigated [14, 15]. Recently, the three-dimensional form of *Sinningia speciosa* was analyzed by micro-CT scanning and generalized Procrustes analysis (GPA) based on a landmark method [16]. These studies showed that three-dimensional quantification can identify flower openings and corolla asymmetry, which was difficult to ascertain with the two-dimensional method, and these morphological traits are useful for illustrating the morphological transition between actinomorphic and zygomorphic flowers [17]. In a subsequent study, Hsu *et al.* [18] investigated the qualitative relationship between morphological traits, such as curvature and tube dilation, and the genotype of the *SsCYC* gene, which is a homolog of the *CYCLOIDEA2*-like gene and is associated with the morphological transition between actinomorphic and zygomorphic flowers. However, the method used for *S. speciosa* is not suitable for quantifying three-dimensional floral morphologies with complex and/or hierarchical structures, because GPA requires morphological homology of landmarks. For example, although corresponding points on individual floral organs can be defined based on the geometrical and biological backgrounds, it is difficult to conserve these corresponding points on the whole floral structure assembled from an indefinite number of floral organs.

In this article, we propose a new theoretical morphological model to quantitatively describe the three-dimensional floral morphology in water lilies (*Nymphaea*) (Fig 1). Water lilies belong to an ancestral group of angiosperms [19], and display some unique features, such as spiral

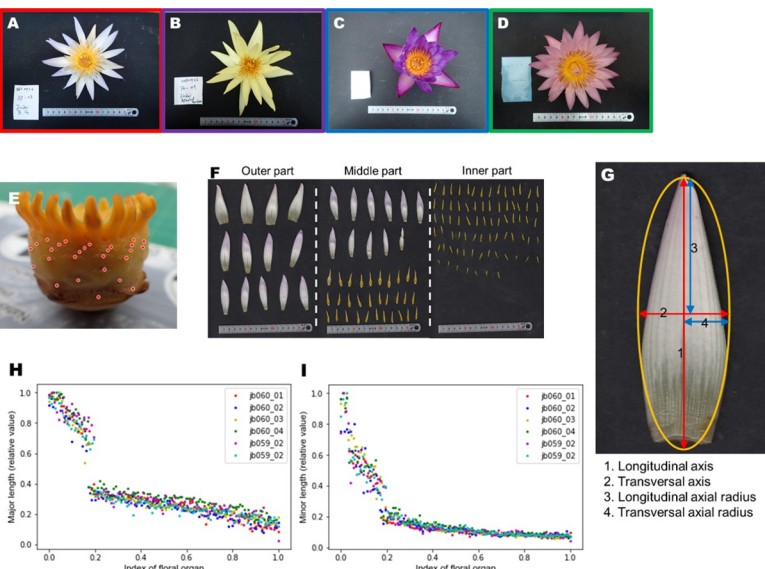

**Fig 1. Observation of floral structure.** (A-D) Whole flower images of (A) *Nymphaea* 'Dauben' ("cup-like"), (B) *Nymphaea* 'Eldorado' ("stellate"), (C) *Nymphaea* 'Lindsey Woods' ("unknown"), and (D) *Nymphaea* 'Pink Platter' ("others"). The frame color of the image corresponds to a flower form: Red: "cup-like", Purple: "stellate", Green: "others", Blue: "unknown". (E) Floral structure without floral organs. The attachment positions of floral organs are marked with red points. (F) Removed floral organs (tepals and stamens). The "outer part", "middle part", and "inner part" indicate relative positions on the ovary (G) Approximation of floral organs to an ellipse and its denomination. (H), (I): Gradual transitions of floral organs of *Nymphaea* 'Dauben' relative to the length of individual flowers. (H) Transverse axial radius, (I) Longitudinal axial radius. The data are normalized to 0–1 as they were relative values. The lateral axis represents the normalized numbers of floral organs.

phyllotaxis, indefinite number of floral organs, and gradual changes in floral organ identities. In addition, there is a change in size and shape from the outer organs (tepals) to the inner organs (stamens) [20], and intermediate organs between the tepals and stamens appear around the region where organ identities switch. Johann Wolfgang von Goethe regarded this transition in serial organs as an example of his "metamorphosis" theory [21]. In the context of modern plant science, this phenomenon has been explained as the "fading border model" [22], one of the expanded ABCE models to explain floral structures of basal angiosperms based on molecular developmental genetics. Water lily is a suitable subject for floral-structure evolution studies [23]. In cultural history, water lilies have been used as food, in traditional medicine, and as religious symbols, especially in Egypt and Asian countries [24], and various water lily cultivars have been developed in western countries since the 19[th] century [25]. Ornamental cultivars show a wide range of floral morphological characteristics, which are partly ascribed to changes in the expression patterns of MADS-box genes, which regulate the floral organ identities from the original species [26]. Recently, the whole-genome sequence of *Nymphaea colorata* was decoded [27] and is expected to improve our understanding of the genetic mechanisms underlying morphological variations in water lilies. Therefore, concise analysis of the water lily floral morphology may provide important information for studies in the field of breeding, genetics, and evolution, including evolutionary developmental (evo-devo) biology [28, 29].

We developed a novel approach to assess the floral morphology in water lily based on a theoretical morphological model [30] consisting of three hierarchical descriptions, i.e., descriptions of (1) the forms of serial floral organs, (2) the phyllotaxis of floral organs, and (3) the opening level of an individual flower. We assessed the silhouettes of flowers generated

theoretically using two geometrical indices, i.e., convexity and solidity. Finally, we discussed the applicability of our model in evolutionary and horticultural studies.

## Materials and methods

In this section, we propose a theoretical morphological model describing the three-dimensional floral morphology in water lily, which has a complex and hierarchical floral structure.

### Theoretical model

Flowers of water lily show spiral phyllotaxis, with both size and shape of floral organs exhibiting a gradual transition. We developed a theoretical model based on three components (Fig 2): (1) shape of floral organs, (2) spatial placement of organs on the ovary, and (3) elevation angles of the organs. We assumed that the size and shape of floral organs in a flower changed in a linear manner. Each floral organ was arranged helically on the ovary in order to express the spiral phyllotaxis. Finally, the opening states of floral organs were defined by elevation angles against the central axis of the flower, which is the axis from the basal center to the apical center of the cylinder approximating the morphology of an ovary.

**The shape of floral organs.**  In this study, every floral organ was approximated as an ellipse, and the transverse and longitudinal axes were parameterized as $x$ and $y$, respectively. We used two piecewise linear functions to represent the gradual transition in both, $x$ and $y$. The former and latter conditions correspond to the sequential transition in tepals and stamens, respectively. Transverse and longitudinal axes of the $i$th organ, $x_i$ and $y_i$, were written as

$$x_i = \begin{cases} x_0 - s_{x1}l_i & (0 \leq i \leq nt_x) \\ x_t - s_{x2}l_i & (nt_x \leq i) \end{cases} \tag{1}$$

$$y_i = \begin{cases} y_0 - s_{y1}l_i & (0 \leq i \leq nt_y) \\ y_t - s_{y2}l_i & (nt_y \leq i) \end{cases} \tag{2}$$

where $x_0$ and $y_0$ are the initial lengths of the transverse and longitudinal axes, respectively (i.e., the length of the outermost organ), $x_t$ and $y_t$ are the lengths of the first organ after the transition thresholds $t_x$ and $t_y$, and $l_i$ is the relative position parameter of the $i$th organ in the total number of organs $n$ ($li = i/n$). Transition parameters placed the border between tepals and stamens, and these parameters function as switches to change the parameters of each organ identity from the tepal to the stamen. We did not assume any developmental mechanism. When $t_x = 1$ or $t_y = 1$, all floral organs become tepals. When $t_x = 0$ and $t_y = 0$, all floral organs become stamens. In the subsequent analyses, we made $t_x = t_y = 0.20$ except in the calculation for Fig 3B. We fixed the total number of floral organs to 100 organs in all analyses, therefore the value of 0.20 gives 20 tepals, a number in the typical range in wild species.

Subscripts "1" and "2" of $s$ in Eqs 1 and 2 indicate the first (tepal) and second (stamen) parts of the piecewise function. The coefficients $s_{x1},s_{x2},s_{y1}$, and $s_{y2}$ represent the steepness, referred to as the "shortening rate" in this study. After an organ reached the transition thresholds $t_x$ and $t_y$, the shortening rate changed from $s_{x1}$ to $s_{x2}$ and from $s_{y1}$ to $s_{y2}$, respectively. We standardized $x_i$ and $y_i$ by the radius $r_b$ of the basal part, i.e., the ovary. The ovary was represented as a cylinder with $r_b = 1$.

**Spiral phyllotaxis.**  Floral organs were arranged around the surface of the ovary, and the arrangement could be expressed as a helix. The arrangement of a floral organ on the ovary was defined by two parameters: the azimuth $\phi$ and the height $h$. The azimuth of the $i$th organ $\phi_i$

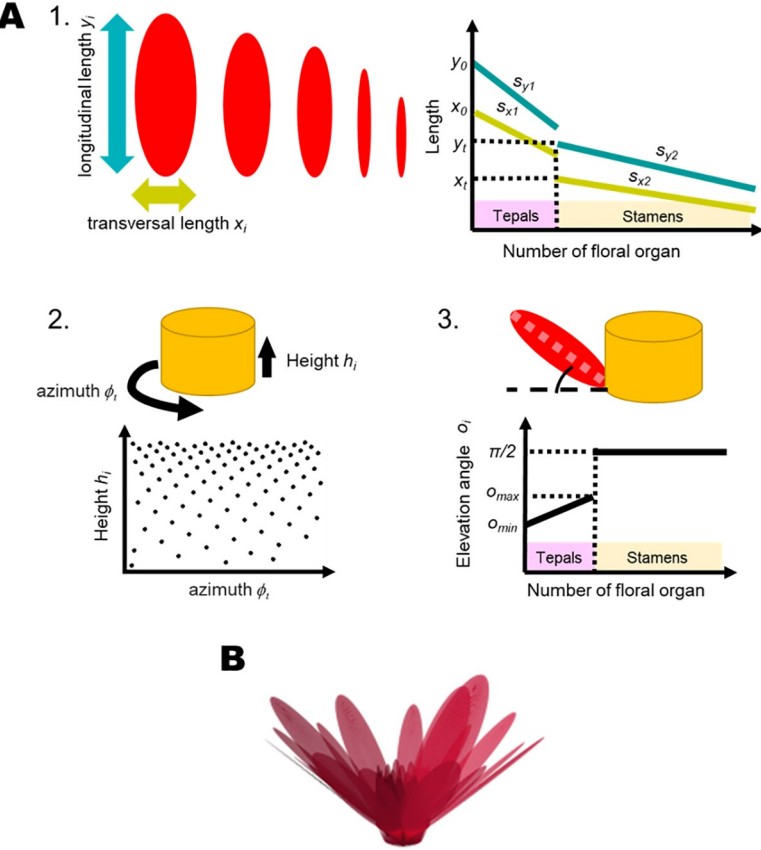

**Fig 2. The three rules applied for morphological modeling.** (A) The three rules are: (1) The gradual transition in sequential organs. Red ellipses illustrate floral organs whose shapes are described by the "longitudinal length" and "transversal length". Blue and yellow lines indicate the transition of organs in the longitudinal lengths and the transversal lengths, respectively. Dotted lines indicate the values of lengths' parameters $x_t$ and $y_t$ at the transition thresholds $t_x$ and $t_y$, reapectively. (2) The spiral phyllotaxis. The yellow cylinder is an ovary from which floral organs detach. Dots in the scatter plot correspond to the azimuths and heights of floral organs. (3) The openness of organs. Black thick lines in the graph represent the elevation angles of floral organs. Dotted lines indicate $\pi/2$ for stamens and elevation angles $o_{max}$ and $o_{min}$ for tepals. (B) A representation of the whole-flower morphology generated by the proposed theoretical morphological model with values presented in Table 1. This floral model does not represent the flower morphology of an actual cultivar.

was given by

$$\varphi_0 = 0$$

$$\varphi_{i+1} = \varphi_i + \Delta\varphi_i$$

$$\Delta\varphi_i = \begin{cases} 90.0° & (i = 1, 2, 3) \\ 137.5° & (i \neq 1, 2, 3) \end{cases} \qquad (3)$$

where the azimuth increases by either 90.0° or 137.5°. Based on some previous studies [31–33], we arranged other organs according to "the golden angle", i.e., ~137.5° (Eq 3). The height was defined as the distance between the organ and the bottom of the ovary. We assumed that each height interval followed an exponential equation. Thus, the height of the $i$+1th organ $h_{i+1}$

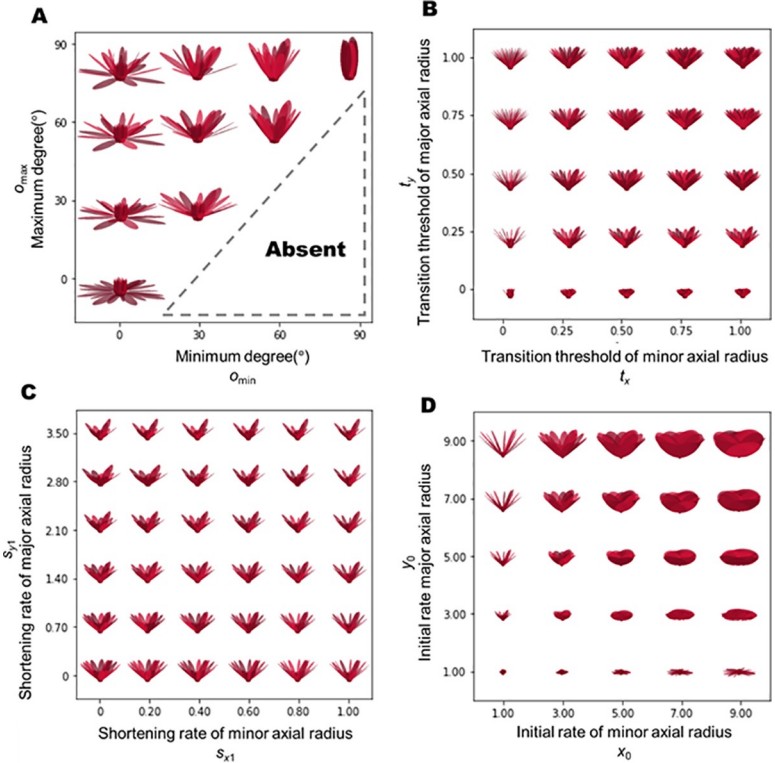

**Fig 3. Effect of parameters on theoretical morphology.** Displayed morphospaces are obtained by modifying the parameters of (A) openness, (B) number of tepals, (C) shortening rate, and (D) initial length.

was given by

$$h_{i+1} = h_i + e^{p\varphi_i}$$

where $p$ is the pitch, which is a constant that regulates the tightness of the intervals. For $i = 0$, we set $h_0 = 0$.

**Elevation angles of the organs.** Floral organs located on the ovary were inclined at their elevation angles $o$. The first 4 tepals had the minimum elevation angle $o_{min}$, whereas the final tepal had the maximum elevation angle $o_{max}$, i.e., $t_{max} = max\,(t_x, t_y)$. The elevation angles of the tepals increased linearly from $o_{min}$ to $o_{max}$. We assumed that all stamens were parallel to the apical-basal axis. The elevation angle of the $i$th organ $o_i$ was given by

$$o_i = \begin{cases} o_{min} & (i = 0, 1, 2, 3) \\ o_{i-1} + \dfrac{o_{max} - o_{min}}{t_{max} - 4} & (4 \leq i \leq t_{max}) \\ \dfrac{\pi}{2} & (t_{max} < i) \end{cases}$$

Although water lily flowers open periodically in the morning or at night (i.e., the appearance of the flower changes during the day), we assumed that elevation angles remained constant in the subsequent analyses for the sake of simplicity. We fixed both the $o_{max}$ and $o_{min}$ at 45° except in the calculation for Fig 3A. However, we did not measure the $o_{max}$ or $o_{min}$ of each flower; the parameters can be adjusted to represent morphological variations in water lilies.

All parameters used in our analysis and their ranges are listed in Table 1.

**Table 1. List of parameters of the theoretical morphological model and their ranges used for the morphospace analysis.**

|  | Explanation | values | ranges | resolutions |
|---|---|---|---|---|
| $n$ | Total numbers of floral organs | 100 | - | - |
| $s_{x1}$ | Shortening rate in "tepals' transverse axis" | 0.80 | 0–1.00 | 0.04 |
| $s_{x2}$ | Shortening rate in "stamens' transverse axis" | 0.02 | - | - |
| $x_0$ | Initial length in "tepals' transverse axis" | 1.00 | 0.50–9.00 | 0.50 |
| $x_t$ | Length in "tepals' transverse axis" after the transition | 0.20 | - | - |
| $t_x$ | Transition threshold in transverse axial length | 0.20 | 0–1.00 | - |
| $s_{y1}$ | Shortening rate in "tepals' longitudinal axis" | 1.40 | 0–3.50 | 0.14 |
| $s_{y2}$ | Shortening rate in "stamens' longitudinal axis" | 0.28 | - | - |
| $y_0$ | Initial length in "tepals' longitudinal axis" | 3.50 | 0.50–9.00 | 0.50 |
| $y_t$ | Length in "tepals' longitudinal axis" after the transition | 1.20 | - | - |
| $t_y$ | Transition threshold in longitudinal axial length | 0.20 | 0–1.00 | - |
| $o_{max}$ | Elevation angle in maximum opening organ | 45˚ | 0˚ - 90˚ | - |
| $o_{min}$ | Elevation angle in minimum opening organ | 45˚ | 0˚ - 90˚ | - |
| $p$ | Pitch of spiral phyllotaxis | -0.01 | $0–10^{-1}$ | - |
| $r_b$ | The radius of the ovary | 1.00 | - | - |
| $h_b$ | The height of the ovary | 0.50 | - | - |

The values are default values, the ranges are parameter ranges, and the resolutions are units in the morphospace analysis.

## Morphospace analysis

We overviewed the theoretical diversity of the floral morphology of water lilies in the morphospace. The morphospace is a parametric space in which each point corresponds to the shape of an individual flower. In this study, four types of morphospaces were assessed: 1) elevation angles, 2) transition thresholds, 3) shortening rates in tepals, and 4) initial lengths.

## Materials

Plants were grown at the Jindai Botanical Gardens (Tokyo, Japan) and the Center for Advanced Biomedical Sciences of Waseda University (Tokyo, Japan). We analyzed 100 specimens (i.e., flowers) belonging to 28 cultivars (Table 2). Floral measurements were recorded in August-November 2016 and May-September 2017.

## Image analysis

Floral organs were detached from each flower starting at the outer and progressing towards the inner position and arranged in the sequence on a flatbed scanner (CanoScan LIDE220, Canon, Tokyo). The order of arrangement of floral organs was checked by a visual inspection. The scanning resolution was 300 dpi. The background was covered with a black paper. We used Fiji, which is a distribution of ImageJ [34], for measuring the axes lengths. First, we binarized the original images and approximated the tepal shape as an ellipse. Then, the longitudinal and transverse lengths were measured using the "Analyze Particles" command in Fiji. The length calibration was based on a scale obtained using scanned floral organs. Initial lengths and shortening rates of sequential organs were calculated for the floral specimens, based on linear regression, using scikit-learn 0.18.1 [35]. The initial length of the transverse axis $x_0$ and longitudinal axis $y_0$ were estimated as

$$x_0 = \frac{b_x}{2}$$

$$y_0 = \frac{b_y}{2}$$

The shortening rates of the transverse axis length $s_x$ and longitudinal axis length $s_y$ were given by

$$s_{x1} = a_x$$

$$s_{y1} = a_y$$

where $a_x$ and $a_y$ are the slopes and $b_x$ and $b_y$ are the intercepts in the linear regression equation. These values were standardized with the radius of each ovary. We assigned every flower to a shape class, as described by Slocum (2005) [36], as follows; "stellate", "cup-like", "other" or "unknown" (Table 2). We treated "star-like" as a synonym of "stellate", and "other" included floral shapes different from both "stellate" and "cup-like". Flowers were categorized as "unknown" when we could not find the description for a cultivar, or when we could not identify the cultivar from the flower.

## Silhouette-based analysis of floral morphology with global feature indices

In this study, theoretical morphologies were expressed as silhouettes projected onto the X-Y plane as the top view, and the Y-Z plane as the side view, to investigate the effects of model parameters on a flower silhouette. The initial lengths and shortening rates were used for this assessment. We calculated the convexity and solidity for quantitative comparison of the theoretical forms [37]. Convexity is the ratio of the perimeters of an object and its convex hull, and can be expressed as

$$C = \frac{L_c}{L_t}$$

where $L_t$ is the perimeter of the target object and $L_c$ is the perimeter of the convex hull. Solidity is the ratio of the area of the convex hull $A_c$ and the area of the object $A_t$

$$S = \frac{A_t}{A_c}$$

OpenCV 3.0.3 [38] was used to calculate these global morphological feature indices.

## Results

### Morphospaces and theoretical models

We obtained morphospaces based on the proposed theoretical model with some parameter ranges (Fig 3). The parameters used in our analysis, and their ranges, are listed in Table 1. The openness of flowers was varied by changing the maximum and minimum elevation angles, $o_{max}$ and $o_{min}$, respectively, (Fig 3A). When $O_{min} = 90°$, the flower was entirely closed.

The theoretical diversity in the morphospace of the transition threshold from tepals to stamens of the transverse ($t_x$) and longitudinal ($t_y$) axes is shown in Fig 3B. In our model, $t_x$ and $t_y$ could be defined independently. The transition thresholds can be used as parameters representing the proportion of the number of tepals in the total number of floral organs. When $t_x$ is equal to $t_y$, the transition from tepals to stamens along both longitudinal and transverse axes

**Table 2. List of plants assessed in this study.**

| Cultivar name | Flower shape [36] | Number of specimens |
|---|---|---|
| Red Cup | Cup-like | 2 |
| Rubra | Stellate | 2 |
| Lindsey Woods | Unknown | 1 |
| Murasaki-Shikibu | Unknown | 3 |
| Marliacea Carnea | Cup-like | 2 |
| Marian Strawn | Stellate | 4 |
| Margaret Randig | Other | 1 |
| White Pearl | Unknown | 11 |
| White Delight | Stellate | 2 |
| Pennsylvania | Stellate | 2 |
| Blue Smoke | Unknown | 3 |
| Blue Indian Goddess | Unknown | 2 |
| Pink Platter | Other | 2 |
| Trailblazer | Unknown | 3 |
| Dauben | Cup-like | 13 |
| Tina | Cup-like | 2 |
| St. Louis Gold | Stellate | 1 |
| General Pershing | Cup-like | 3 |
| Colorata | Cup-like | 8 |
| Queen of Siam | Unknown | 1 |
| Capensis var. Zanzibariensis | Stellate | 4 |
| Enchantment | Other | 4 |
| Eldorado | Stellate | 2 |
| Independence | Unknown | 1 |
| Albert Greenberg | Cup-like | 4 |
| Afterglow | Other | 1 |
| M. E. Whitaker | Stellate | 3 |
| M. E. Hutchings | Unknown | 1 |
| unknown cultivar #1 | Unknown | 1 |
| unknown cultivar #2 | Unknown | 3 |
| unknown cultivar #3 | Unknown | 1 |
| unknown cultivar #4 | Unknown | 2 |
| unknown cultivar #5 | Unknown | 1 |
| unknown cultivar #6 | Unknown | 3 |
| unknown cultivar #7 | Unknown | 1 |

begins at the same floral organ. We found that thin tepals occurred in flowers when $t_x = 0$. Conversely, round tepals were observed when $t_y = 0$.

The theoretical diversity in the morphospace of shortening rate along the transverse ($s_{x1}$) and longitudinal ($s_{y1}$) axes is shown in Fig 3C. Cross-shaped flowers were observed in the theoretical morphospace with high $s_{y1}$ because the outer tepals were noticeable. As $s_{x1}$ increased, interval spaces between the petals became wider, as the tepal width narrowed rapidly relative to the rotation angle of the phyllotaxis ($\phi$). In the morphospace of initial lengths $x_0, y_0$ (Fig 3D), tepals often appeared to overlap with each other, especially in the region satisfying $x_0 > y_0$. Thus, flowers with large diameters were observed with large $y_0$, but not with large $x_0$. When $x_0 = y_0$, flowers appeared to have round tepals.

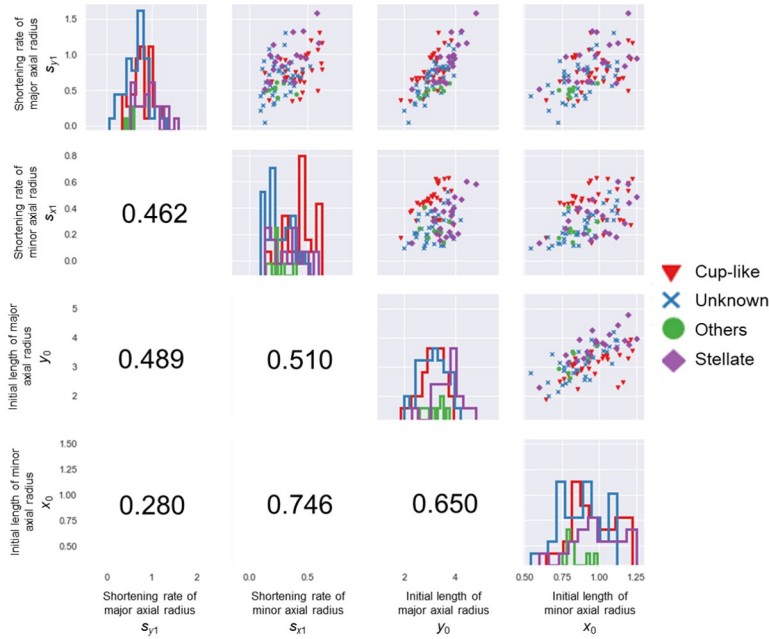

**Fig 4. Pairs plot of the measured values.** Upper diagonal: scatter plots for each combination of parameters. Lower diagonal: Pearson's correlation coefficients. Diagonal: histograms of each parameter. The shape and color of a marker corresponds to a flower form. Red triangle: "cup-like", purple diamond: "stellate", Green circle: "others", blue cross: "unknown". The colors of the histogram are the same as the marker colors.

## Morphometric analysis of water lily flowers

The pairs plot of initial lengths $x_0, y_0$ and shortening rates $s_{x1}, s_{y1}$ is shown in Fig 4. All correlation coefficients were less than 0.8. The measured parameters are summarized in Table 3. The length of the longitudinal axis $y_0$ was greater than that of the transverse axis $x_0$. Thus, initial tepals were always long rather than broad in our measurements. We found that the ratio of the shortening rates ($s_{y1}/s_{x1}$) of "stellate" flowers tended to be larger than that of "cup-like" flowers, and the lengths of tepals shortened easily in "stellate" flowers. In addition, the initial lengths $y_0$ of "stellate" flowers were greater than those of "cup-like" flowers when the value of $s_{y1}$ is same in many cases.

On the contrary, the $s_{x1}$ values of "cup-like" flowers were relatively larger than those of "stellate" flowers in many cases. The aspect ratio of initial lengths ($y_0/x_0$) tended to be larger for "stellate" flowers than for "cup-like" flowers. This tendency suggested that the initial tepals of "stellate" flowers were narrower than those of "cup-like" flowers.

## Evaluation of corolla shape based on silhouette image with convex hull

To evaluate the global shape feature of flowers, we calculated the solidity and convexity of flower silhouettes using different initial lengths $x_0, y_0$ (Fig 5) and shortening rates $s_{x1}, s_{y1}$ (Fig 6). In top-viewed silhouettes, both solidity and convexity were nearly equal to 1 when the aspect ratio of the initial tepal ($x_0/y_0$) was close to 1 (Fig 5C and 5D). Silhouettes in the side view had high solidity when initial lengths of both transverse axis $x_0$ and longitudinal axis $y_0$ were large (Fig 5E). The convexity of side view silhouettes showed a similar pattern, as shown in Fig 5E, except for high values in areas with small $y_0$ (Fig 5F). Thus, all measured $x_0$ and $y_0$ exhibited a similar gradation in each plot. Silhouettes with high solidity and convexity exhibited a round shape in the top view (Fig 5A) and a semicircular shape in the side view (Fig 5B). In the top

**Table 3. Summary of measured parameters.**

| Flower shape | | $s_{y1}$ | $s_{x1}$ | $y_0$ | $x_0$ |
|---|---|---|---|---|---|
| Stellate (n = 20) | Mean | 0.919 | 0.328 | 3.714 | 0.986 |
| | Std. | 0.286 | 0.140 | 0.531 | 0.166 |
| | Min | 0.516 | 0.134 | 2.285 | 0.595 |
| | Max | 1.588 | 0.600 | 4.784 | 1.249 |
| Cup (n = 34) | Mean | 0.780 | 0.430 | 3.080 | 0.962 |
| | Std. | 0.236 | 0.130 | 0.469 | 0.151 |
| | Min | 0.342 | 0.133 | 1.859 | 0.638 |
| | Max | 1.300 | 0.627 | 3.933 | 1.225 |
| Other (n = 8) | Mean | 0.508 | 0.241 | 3.194 | 0.835 |
| | Std. | 0.074 | 0.081 | 0.399 | 0.084 |
| | Min | 0.390 | 0.140 | 2.597 | 0.726 |
| | Max | 0.607 | 0.404 | 3.761 | 0.989 |
| Unknown (n = 38) | Mean | 0.656 | 0.255 | 3.110 | 0.877 |
| | Std. | 0.265 | 0.111 | 0.539 | 0.146 |
| | Min | 0.048 | 0.092 | 1.980 | 0.537 |
| | Max | 1.310 | 0.525 | 4.192 | 1.121 |
| Total (n = 100) | Mean | 0.739 | 0.328 | 3.227 | 0.924 |
| | Std. | 0.274 | 0.144 | 0.555 | 0.155 |
| | Min | 0.048 | 0.092 | 1.859 | 0.537 |
| | Max | 1.588 | 0.627 | 4.784 | 1.249 |

view, silhouettes with deeper notches had lower values for both solidity and convexity. Such deep notches resulted from a high aspect ratio of tepals, i.e., high $y_0$ or high $x_0$.

When the shortening rates $s_{x1}$ and $s_{y1}$ were small, both solidity and convexity were high, as most tepals had a similar sizes and shapes, and these tepals filled the silhouettes in each view (Fig 6). In the top view, the solidity decreased with increasing $s_{x1}$ and $s_{y1}$ (Fig 6C). In general, the -convexity decreased as $s_{x1}$ increased, while the convexity increased with increasing $s_{y1}$ (Fig 6D). The solidity of the side-viewed silhouettes decreased as $s_{y1}$ increased, however it was almost independent of $s_{x1}$ (Fig 6E). The convexity of side-viewed silhouettes decreased as $s_{x1}$ and $s_{y1}$ increased, except when $s_{y1}$ was low (Fig 6F). In both views, the smallest convexity values were obtained in plots with $s_{x1} = 1$ and $s_{y1} = 0$. Flowers with high $s_{y1}$ had low solidity in side-viewed silhouettes, in addition to flowers with small values for both indices in the top view. Each combination of the index and the view revealed a different pattern.

## Discussion

### The theoretical model of water lily flowers and scope of its applicability

Our proposed model, though simple, can effectively describe various flower shapes in water lily (Figs 2 and 3). It can be used to generate a spectrum of floral morphology based on openness (Fig 3A), the proportion of the number of tepals (Fig 3B), and measurable parameters in the gradual transition of floral organs (Fig 3C and 3D). Water lilies show various opening states depending on the species and cultivars (Fig 3A). In horticultural catalogs, such as "Water lilies and lotuses: species, cultivars, and new hybrids" [36], the openness of the flower is categorized as either, "flat" "wide open" or "plate". Although it was difficult to find a one-to-one correspondence between the simulated and real flowers, it appeared that these morphological models could represent possible opening states. For example, flowers of species belonging to subgenus *Lotos*, such as *Nymphaea lotus*, can be generated with $o_{max} = o_{min} = 0°$. Similarly,

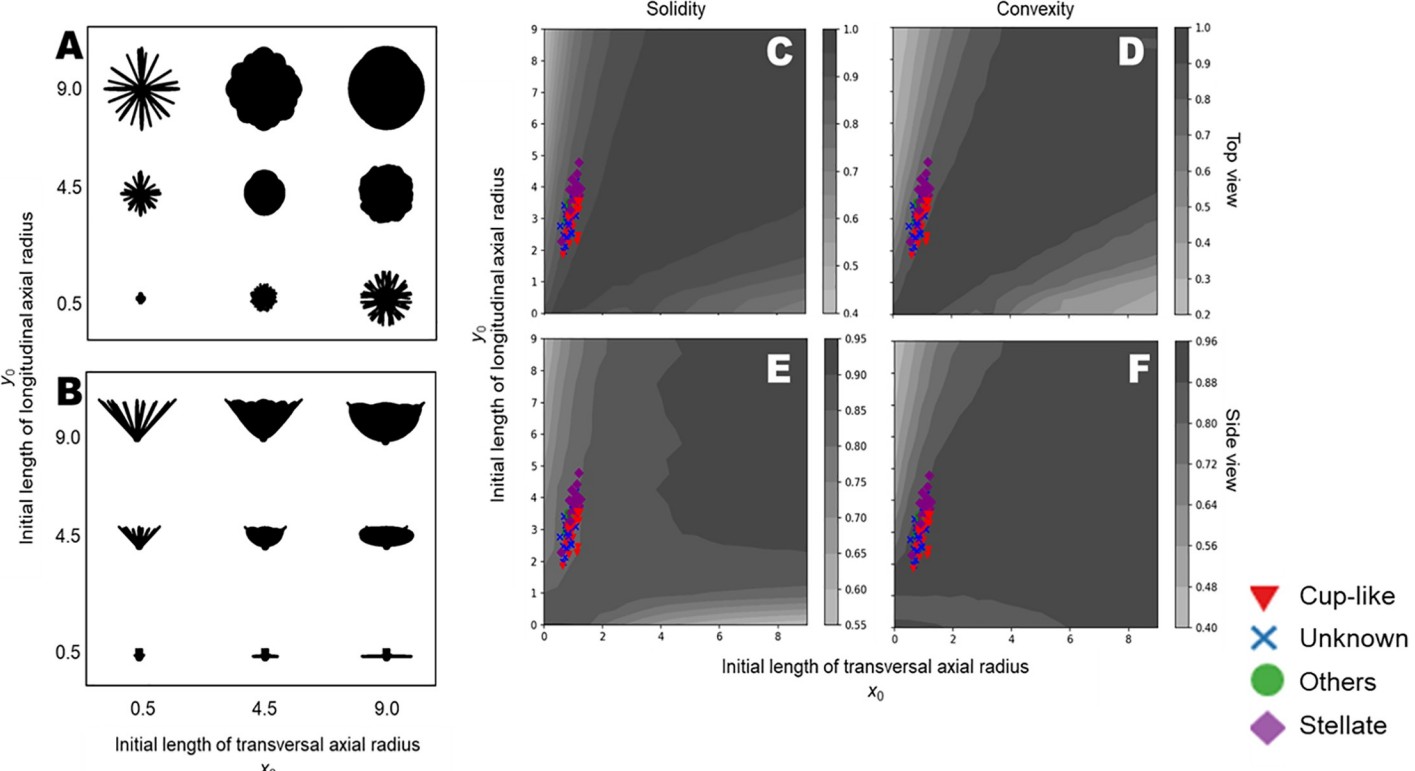

**Fig 5. Assessment of the initial length using the convex hull.** (A) Top view silhouettes, (B) side view silhouettes, (C) distribution of solidity in top view, (D) distribution of convexity in top view, (E) distribution of solidity in side view, (F) distribution of convexity in side view. Colors indicate the relative value of each index. The points are calculated values of solidity or convexity of the respective flowers. Shapes and colors of the points represent the flower forms. Red triangles, purple diamonds, green circles, and blue crosses represent "cup-like", "stellate", "others" and "unknown", respectively.

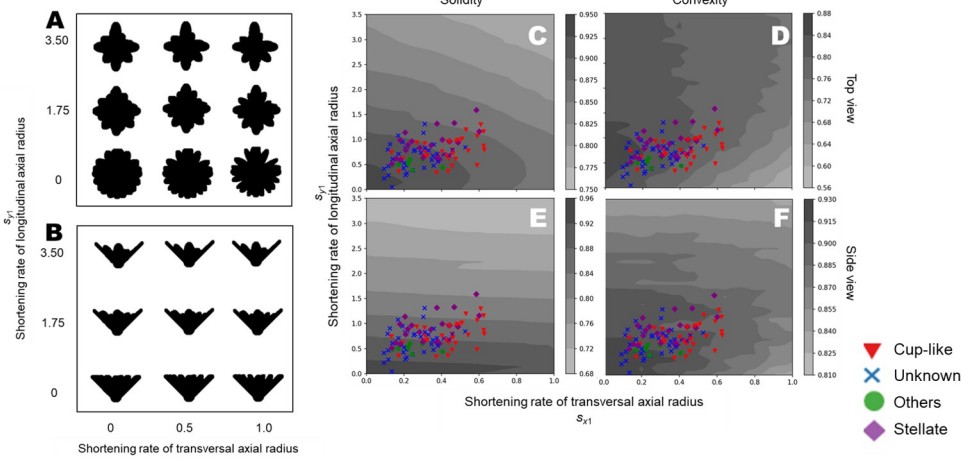

**Fig 6. Assessment of the shortening rate using the convex hull.** (A) Top view silhouettes, (B) side view silhouettes, (C) distribution of solidity in top view, (D) distribution of convexity in top view, (E) distribution of solidity in side view, (F) distribution of convexity in side view. Colors indicate the relative value of each index. The points are calculated values of solidity or convexity of the respective flowers. Shapes and colors of the points represent the flower forms. Red triangles, purple diamonds, green circles and blue crosses represent "cup-like", "stellate", "others" and "unknown", respectively.

the flower of *Nymphaea caerulea*, belonging to subgenus *Brachyceras*, can be generated with $o_{max} = o_{\min} = 60°$. Flowers in subgenus *Nymphaea*, such as *Nymphaea tuberosa* or *Nymphaea odorata*, may show diverse combinations of $o_{max}$ and $o_{min}$. It must be noted that openness also depends on the diurnal movement of tepals.

We attempted to find a correspondence between flowers of real cultivars and theoretical flowers in the morphospace of transition thresholds $t_x$ and $t_y$ (Fig 3B). Flowers of some ornamental cultivars have numerous tepals but are without stamens, and are known as "doubled" flowers. Our model could express such doubled flowers when $t_x \approx 1$ or $t_y \approx 1$ (i.e., almost all floral organs were regarded as tepals) (Fig 3B). Transition thresholds from tepals to stamens $t_x, t_y$ are important for describing not only the double flowers, but also the "full" flowers [36], because flowers containing many tepals can appear by tuning the openness. Flowers with more tepals than stamens resemble double flowers, such as *Nymphaea* 'King of Siam' or *Nymphaea* 'Midnight' (Fig 3B). We did not find any flowers with $t_x = 0$ or $t_y = 0$ in the cultivars assessed in this study.

The shortening rate of the transverse axis $s_{x1}$ of some real species and cultivars was observed to be between 0.091 and 0.627 (Figs 3C and 4). Conversely, the shortening rate of the longitudinal axis $s_{y1}$ of real cultivars was found to be between 0.048 and 1.588. We found some flowers in which the four outermost tepals were relatively larger than other tepals, which we referred to as "cross-shaped" flowers, although such variations have not been reported in breeder's catalogs.

The theoretical model yielded unrealistic morphologies in a large part of the morphospace of $x_0$ and $y_0$ (Fig 3D). When $x_0 = y_0$, the tepals were rounded. A round tepal is rarely found in *Nymphaea*, but is typical to *Nuphar*, another genus in family Nymphaeaceae. In the region where $x_0 > y_0$, cultivars with overlapping tepals in real flowers can be difficult to find, because such a morphology may require intricate folding, which may be unsuitable for periodic opening and closing of the flower. Moreover, no real cultivars have been reported to have very large tepals. This implies that flowers with very large $x_0$ and $y_0$ relative to the ovary do not occur easily due to both functional and developmental constraints, as such flowers would not be able to fold up their tepals even if the arrangement was physically possible. We found that the assessed flowers always had long, rather than broad, first tepals.

In our model, we could not control the acuteness of the tepal tip, wavy surface, and curly form. Therefore, it was difficult to express some textures observed in species like *Nymphaea gigantea* or *Nymphaea* 'Gloire du Temple-sur-Lot'. However, the theoretical morphological approach can be useful for researchers and breeders to describe and understand the complicated relationships between floral organs in an entire individual flower. The integration of hierarchical floral organs allows us to construct a three-dimensional structure from easily measurable elements. Our approach can facilitate the three-dimensional analysis of floral morphology by employing a combination of two-dimensional image-based simple measurements and a theoretical morphological model.

## Observed data and model structure

We classified the measurement data according to the flower form descriptions from a breeder's catalog, as shown in Fig 4. The "stellate" character, which is one of the most common flower shapes, is associated with the tepal width. The initial tepal of a "stellate" flower tends to be longer and narrower than that of a "cup-like" flower. As seen in Fig 5, "stellate" flowers tended to be in a region with relatively high $y_0$. In the catalog [36], several cases of "broad petal" and "narrow petal" were reported, and these characters may be expressed by changing the $s_{x1}$. It indicated that tepals in a "stellate" flower can quickly transition to shorter and rounder forms,

whereas those in a "cup-like" flower tend to transition to a narrower shape. It is possible that interval spaces could be easily generated in "stellate" flowers, and that the overlapping of tepals resulted in a round-shaped silhouette of "cup-like" flowers in our observations. Thus, we can conclude that in order to estimate the three-dimensional morphology, it is essential to describe not only the length and width of each tepal, but to also capture those transitions.

## Characteristics of the global feature indices of a flower silhouette

We calculated two global feature indices, i.e., solidity and convexity, of the simulated flower silhouettes, and investigated the relationship between two types of parameters, i.e., initial lengths $x_0, y_0$ and shortening rates $s_{x1}$, $s_{y1}$, and solidity and convexity. Solidity is defined as the ratio of the silhouette area to its convex hull, while convexity is defined as the ratio of the perimeters of the silhouette to its convex hull. Convexity can reflect the depth of the notches, while there is no significant effect of depth on solidity, suggesting that convexity is suitable for evaluating the sparseness of tepals (e.g., size and frequency of notches between tepals), and solidity describes the level of extension of tepals.

The silhouette of a simulated flower and its global shape indices allow us to identify global shape changes through changes in the parameters of the model. For example, a change in $s_{x1}$ alters the convexity of the top view silhouette, while a change in $s_{y1}$ does not (Fig 6D). Similarly, a change in $s_{x1}$ does not alter the solidity of the side view silhouette, while a change in $s_{y1}$ does (Fig 6E). It is not always easy to understand the influence of theoretical parameters on the global morphological features of a flower. The approaches demonstrated in this study can provide a useful way to translate the variations in global morphological features into variations in theoretical morphological parameters. With this approach, we may be able to find the theoretical morphological parameter values required for obtaining a flower with desirable global features.

## Summary and possible applications

In this paper, we proposed a theoretical morphological model that represents the three-dimensional floral morphology of *Nymphaea*, and evaluated the flower silhouettes using global feature indices defined using the convex hull. The model was able to represent the general floral morphologies of genus *Nymphaea*. In addition, we noted that it was important to capture the gradual transition of tepals to describe an individual flower, and we estimated sequences of the gradual transition of tepal forms in real specimens based on our theoretical model. Our model may be able to contribute to evo-devo studies in botany because the fading border model [22] assumes a gradual change in organ characteristics, which can be captured by our model as the gradual transition of organ shape. Further generalization and formalization are necessary to use our model for morphogenetic analyses. For example, we modeled the shape transition with a simple linear function, however, more complex functions may be more appropriate for such morphogenetic analyses.

In our model, we evaluated the silhouettes using solidity and convexity, both of which were defined in relation to the convex hull. Although these indices were relatively simple measurements, they were able to effectively capture the global morphological features. Our approach, combining the theoretical morphological model and silhouette evaluation, can clarify the relationship between the qualitative description of floral morphology in a horticultural catalog and the quantitative shape variations captured by our model.

In this paper, we also assessed the association between the estimated parameters and flower shape descriptions. Categorical descriptions, though simple and intuitive, are often too subjective and require expert knowledge. The approach based on a theoretical model, as proposed in

this study, may allow us to translate the categorical descriptions provided by breeders to qualitative expressions, and to overview the semantic structures of complex phenotypic varieties. We expect that this approach will facilitate the bridging of horticultural archives to other fields in plant sciences.

## Supporting information

**S1 Striking image.**
(TIF)

## Acknowledgments

We acknowledge and appreciate the kind cooperation of the Jindai Botanical Gardens (Tokyo, Japan), for providing the biological materials for this study in 2016–2017. We thank members of metaPhorest (bioaesthetics platform, Tokyo) for encouragements and discussion.

## Author Contributions

**Conceptualization:** Koji Noshita.

**Investigation:** Shiryu Kirie.

**Methodology:** Shiryu Kirie, Koji Noshita.

**Supervision:** Hideo Iwasaki, Koji Noshita, Hiroyoshi Iwata.

**Writing – original draft:** Shiryu Kirie.

**Writing – review & editing:** Koji Noshita, Hiroyoshi Iwata.

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
