## [Decision Letter · Decision Letter 0]

20 Aug 2020

PONE-D-20-13840

A theoretical morphological model for quantitative description of the three-dimensional floral morphology in waterlily (Nymphaea)

PLOS ONE

Dear Dr. Iwata,

Thank you for submitting your manuscript to PLOS ONE. After careful consideration, we feel that it has merit but does not fully meet PLOS ONE’s publication criteria as it currently stands. Therefore, we invite you to submit a revised version of the manuscript that addresses the points raised during the review process.

This manuscript is providing a new method to characterize waterlily flowers which is important for understanding the diversification of basal angiosperms. By introducing this method, the importance of this species in evolutionally biology will be increased. However, as mentioned by a reviewer, it needs to improve the description of developed method in detail for audiences of this journal.  

We look forward to receiving your revised manuscript.

Kind regards,

Hiroshi Ezura

Academic Editor

PLOS ONE

Journal Requirements:

Reviewers' comments:

Reviewer's Responses to Questions

**Comments to the Author**

1. Is the manuscript technically sound, and do the data support the conclusions?

Reviewer #1: Yes

2. Has the statistical analysis been performed appropriately and rigorously? 

Reviewer #1: I Don't Know

3. Have the authors made all data underlying the findings in their manuscript fully available?

Reviewer #1: Yes

4. Is the manuscript presented in an intelligible fashion and written in standard English?

Reviewer #1: Yes

5. Review Comments to the Author

Reviewer #1: The authors constructed a theoretical model of three-dimensional morphology of waterlily using simple components. The method can generate three-dimensional representation of waterlily flowers. Large part of morphological variations of waterlily can be displayed using the method with varying parameters. The authors also found a relation between some parameters and shape descriptions from flower breeders. The method is sufficiently novel, and this manuscript will provide an important contribution to the field of research. However, I feel that some corrections are still necessary.

I feel there are insufficient explanation in the Material and Methods section. I would suggest that the authors include enough information with appropriate references and programs used in this study, such that an independent investigator would be able to repeat the experiments.

Figure legends need further description. A handbook says that "A legend is needed so that the figure will be intelligible without reference to the text" (Zaiger, 2000, Essentials of Writing Biomedical Research Papers).

The URL the authors mentioned the data of the manuscript (https://github.com/ShiryuKirie/theoretical-morphological-model-of-water-lily) is dead link. Please could you show the right URL?

I also put comments in the word file attached.

I hope these comments would be helpful.

6. PLOS authors have the option to publish the peer review history of their article (what does this mean?). If published, this will include your full peer review and any attached files.

Reviewer #1: No

---

## [Author Response · Author response to Decision Letter 0]

10 Sep 2020

Reviewer #1: The authors constructed a theoretical model of three-dimensional morphology of waterlily using simple components. The method can generate three-dimensional representation of waterlily flowers. Large part of morphological variations of waterlily can be displayed using the method with varying parameters. The authors also found a relation between some parameters and shape descriptions from flower breeders. The method is sufficiently novel, and this manuscript will provide an important contribution to the field of research. However, I feel that some corrections are still necessary.

We are glad about your understanding and agree with your suggestion that this paper should improve in the explain of Materials and Methods. Your comments have brought us good perspectives in our revision, as we would propose below.

I feel there are insufficient explanation in the Material and Methods section. I would suggest that the authors include enough information with appropriate references and programs used in this study, such that an independent investigator would be able to repeat the experiments.

We appreciate your suggestions. As you suggested, the original manuscript was not sufficient to understand how to calculate the theoretical model and to repeat the experiments. 

We added the explanation on how to get scanned images in the line 254-257 and line 261-262. 

In line 181-188, we wrote what the transition parameters are and how to give them to the model. In almost all analyses, we fixed these parameters for the sake of simplicity.

We also explained how to judge the opening state in line 225-229. Actually, we did not judge the opening because the opening state depends highly on the environment and the condition of observation. We fixed parameters of elevation angle in many analyses and intend to facilitate the comparison among the different morphologies.

Figure legends need further description. A handbook says that "A legend is needed so that the figure will be intelligible without reference to the text" (Zaiger, 2000, Essentials of Writing Biomedical Research Papers).

We are sorry that some figures were not easy to read without the text. In this revision, we have rewritten the legend to make it comprehensive. We will explain how we modified figures and those legends.

Fig 1:

We provided three additional pictures of flowers and added the information each flower forms.

Figs 2:

We modified Fig 2A to improve the readability and added the explanation on those figures. Additionally, we change the Fig 2B into the representation with parameters in list 1.

Figs 4:

We made the legend of markers bigger.

Figs 5, Figs 6:

We added the legend of markers.

Line 27, 31: "waterlily"?

We changed it to use “water lily” in all part of this paper including the title following an advice from English proofreading service.

Line 68-71: Would you cite appropriate references?

The contents of this sentence are written in the reference [9], therefore we had added the reference (Line68). 

Line 102-103: I feel that it would be an overstatement to say that. I would suggest that the language would be softened, for example, " Waterlilies are suitable materials for evolutionary studies of floral structure".

I completely agree to the suggestion. We revised the sentence as you suggested (Line 103-104). 

Line 115: Would you show the cultivar name of the flower? Is it typical "stellate"? In addition, I recommend putting pictures of various types of flower shapes, "stellate" "cup-like", "other" or "unknown".

We completely agree with your suggestion. We have added pictures of other 3 types of flowers in Fig 2A-D and revised its legend as

(A - D) Whole flower images of (A) Nymphaea ‘Dauben’ (“cup-like”), (B) Nymphaea ‘Eldorado’ (“stellate”), (C) Nymphaea ‘Lindsey Woods’ (“unknown”), and (D) Nymphaea ‘Pink Platter’ (“others”). The frame color of the image corresponds to a flower form: Red: “cup-like”, Purple: “stellate”, Green: “others”，Blue: “unknown”. (Line 118-121)

The alphabets denoting the panels were also modified according to this change. For example, (B) in the original version was changed as (E), (C) as (F), and so on.

Additionally, we added the following sentence to explain (F):

The "outer part", "middle part", and "inner part" indicate relative positions on the ovary (Line 123-124)

Line 116: How did you classify "outer part", "middle part" and "inner part"? Would you describe in Materials and Methods section?

Organs are detached from a flower in turn and just arranged in the sequence on the sheets. The reason why we used "outer part", "middle part" and "inner part" is for convenience in the display. Places of organ on a scanned image reflected an order of organs located on the ovary. 

In Materials and Methods, we added the explanation for the arrangement of sheets described above as,

Floral organs were detached from each flower starting at the outer and progressing towards the inner position and arranged in the sequence on a flatbed scanner (CanoScan LIDE220, Canon, Tokyo). The order of arrangement of floral organs was checked by a visual inspection. (Line 254-257)

And we wrote additional information in the measurement,

The length calibration was based on a scale obtained using scanned floral organs.(Line 261-262)

Line148: I would suggest that the authors add the description for the graph in Fig 2. What was the dotted line vertical to Y-axis?

We are sorry for the insufficient descriptions. Dotted lines indicated the values of lengths parameters x_t and y_t at the transition thresholds in Fig 2A (1) or elevation angles o_max and o_minin Fig 2A (3). We have added sentences for explaining Fig 2 as

(A) The three rules are: (1) The gradual transition in sequential organs. Red ellipses illustrate floral organs whose shapes are described by the “longitudinal length” and “transversal length”. Blue and yellow lines indicate the transition of organs in the longitudinal lengths and the transversal lengths, respectively. Dotted lines indicate the values of lengths’ parameters x_t and y_t at the transition thresholds t_x and t_y, reapectively. (2) The spiral phyllotaxis. The yellow cylinder is an ovary from which floral organs detach. Dots in the scatter plot correspond to the azimuths and heights of floral organs. (3) The openness of organs. Black thick lines in the graph represent the elevation angles of floral organs. Dotted lines indicate π/2 for stamens and elevation angles o_max and o_min for tepals. … (Line 155-164)

Line149-150: Which cultivar was used to generate the representation of the whole-flower morphology? Would you describe the detailed information, such as specific parameter values? Also I recommend you add a photo of actual flower of the cultivar you used for modeling side-by-side.

This representation is a kind of “default” flower morphology based on the parameter values listed in Table1. We have explained this point in the legend of Fig 2B as

(B) A representation of the whole-flower morphology generated by the proposed theoretical morphological model with values presented in Table 1. This floral model does not represent the flower morphology of an actual cultivar. (Line 165-167)

Line163: It would be useful to include enough information for the transition thresholds Tx and Ty. How were these parameters given?

Line 387: This information should be also added in Materials and Methods section. 

These parameters are arbitrary parameters to switch the effects of “organ identity” from the tepal to the stamen. We did not assume any developmental mechanism, but this setting enhances the flexibility of our model (i.e., the model can be flexibly adapted to various types of morphological variations). In the subsequence analyses, we set Tx = Ty (=0.20) except Fig3B, and it means the change in the effect of organ identity starts at the same organ. The value 0.20 gives 20 tepals because the total number of floral organs are fixed to 100 organs in this paper. These are numbers of tepals or total organs in the typical ranges in the wild species

In Line 181-188, we have added sentences explaining this point as

Transition parameters placed the border between tepals and stamens, and these parameters function as switches to change the parameters of each organ identity from the tepal to the stamen. We did not assume any developmental mechanism. When t_x=1 or t_y=1, all floral organs become tepals. When t_x=0 and t_y=0, all floral organs become stamens. In the subsequent analyses, we made t_x = t_y = 0.20 except in the calculation for Fig 3B. We fixed the total number of floral organs to 100 organs in all analyses, therefore the value of 0.20 gives 20 tepals, a number in the typical range in wild species.(Line 181-188)

Line193-194: It was difficult for me to follow how to calculate Omin and Omax. Did you actually measure these angles? Would you describe how to calculate Omin and Omax in detail? 

We did not measure Omin and Omax themselves in this study, but just assumed their values. The flower of waterlily shows different levels of openness in different times (it is always fluctuating and does not have fixed angles). Because of the variability of the angles, we assumed that they were fixed as Omin = Omax = 45° for the unification of the opening condition and the simplification.

Line 200-202: How did you judge the flower entirely opened?

We did not judge the entire open of a flower because the openness of the flower was not measured. We measured only lengths and widths of tepals and used the measured values in the model. In Line 225-229, we added sentences explaining this point as

…, we assumed that elevation angles remained constant in the subsequent analyses for the sake of simplicity. We fixed both the o_max and o_min at 45° except in the calculation for Fig 3A. However, we did not measure the o_max or o_min of each flower; the parameters can be adjusted to represent morphological variations in water lilies. (Line 225-229)

Line 224: I recommend putting information on flower shape (stellate, cup-like, other or unknown) in each cultivar.

Your suggestion will enhance the information of cultivars. We added information about the types of flower shape in the Table 2.

Line 287-288: This sentence should correspond to result section.

We tried modifying the expression and combined to preceding sentence like;

… The transition thresholds can be used as parameters representing the proportion of the number of tepals in the total number of floral organs. (Line 315-317)

Line 357: I cannot understand what the meaning of this part.

We are sorry that the original sentence had an error. I corrected it as

… the convexity increased with increasing s_y1 (Line 392)

Line 346, 350: Information of the shape and color of a marker should be described even if it was same as in Fig 4.

Thank you for the suggestion. We followed your suggestion and added necessary information to the legends of Figs 5 and 6:

Shapes and colors of the points represent the flower forms. Red triangles, purple diamonds, green circles, and blue crosses represent “cup-like”, “stellate”, “others” and “unknown”, respectively. (Line 377-379 and Line 384-386)

Line 570: Would you provide a reference in English? Or, please indicate clearly the reference written in French.

We found English edition, so we substituted the reference.

Holmes C. Water Lilies and Bory Latour-Marliac, the Genius Behind Monet's Water Lilies. Garden Art Press; 2015 (Line 607-608)

The URL the authors mentioned the data of the manuscript (https://github.com/ShiryuKirie/theoretical-morphological-model-of-water-lily) is dead link. Please could you show the right URL?

We changed the visibility from “private” to “public”.

Could you check the same URL again?

---

## [Editor Report · Decision Letter 1]

14 Sep 2020

A theoretical morphological model for quantitative description of the three-dimensional floral morphology in water lily (Nymphaea)

PONE-D-20-13840R1

Dear Dr. Iwata,

We’re pleased to inform you that your manuscript has been judged scientifically suitable for publication and will be formally accepted for publication once it meets all outstanding technical requirements.

Kind regards,

Hiroshi Ezura

Academic Editor

PLOS ONE
---

## [Editor Report · Acceptance letter]

1 Oct 2020

PONE-D-20-13840R1 

A theoretical morphological model for quantitative description of the three-dimensional floral morphology in water lily (*Nymphaea*) 

Dear Dr. Iwata:

I'm pleased to inform you that your manuscript has been deemed suitable for publication in PLOS ONE. Congratulations! Your manuscript is now with our production department. 

Kind regards, 

on behalf of

Prof. Hiroshi Ezura 

Academic Editor

PLOS ONE